# Resurrecting the sigmoid in deep learning through dynamical isometry: theory and practice

**Jeffrey Pennington**
Google Brain

**Samuel S. Schoenholz**
Google Brain

**Surya Ganguli**
Applied Physics, Stanford University and Google Brain

## Abstract

It is well known that weight initialization in deep networks can have a dramatic impact on learning speed. For example, ensuring the mean squared singular value of a network's input-output Jacobian is $O(1)$ is essential for avoiding exponentially vanishing or exploding gradients. Moreover, in deep linear networks, ensuring that *all* singular values of the Jacobian are concentrated near 1 can yield a dramatic additional speed-up in learning; this is a property known as dynamical isometry. However, it is unclear how to achieve dynamical isometry in nonlinear deep networks. We address this question by employing powerful tools from free probability theory to analytically compute the *entire* singular value distribution of a deep network's input-output Jacobian. We explore the dependence of the singular value distribution on the depth of the network, the weight initialization, and the choice of nonlinearity. Intriguingly, we find that ReLU networks are incapable of dynamical isometry. On the other hand, sigmoidal networks can achieve isometry, but only with orthogonal weight initialization. Moreover, we demonstrate empirically that deep nonlinear networks achieving dynamical isometry learn orders of magnitude faster than networks that do not. Indeed, we show that properly-initialized deep sigmoidal networks consistently outperform deep ReLU networks. Overall, our analysis reveals that controlling the *entire* distribution of Jacobian singular values is an important design consideration in deep learning.

## 1 Introduction

Deep learning has achieved state-of-the-art performance in many domains, including computer vision [1], machine translation [2], human games [3], education [4], and neurobiological modelling [5, 6]. A major determinant of success in training deep networks lies in appropriately choosing the initial weights. Indeed the very genesis of deep learning rested upon the initial observation that unsupervised pre-training provides a good set of initial weights for subsequent fine-tuning through backpropagation [7]. Moreover, seminal work in deep learning suggested that appropriately-scaled Gaussian weights can prevent gradients from exploding or vanishing exponentially [8], a condition that has been found to be necessary to achieve reasonable learning speeds [9].

These random weight initializations were primarily driven by the principle that the mean squared singular value of a deep network's Jacobian from input to output should remain close to 1. This condition implies that on *average*, a randomly chosen error vector will preserve its norm under backpropagation; however, it provides no guarantees on the worst case growth or shrinkage of an error vector. A stronger requirement one might demand is that *every* Jacobian singular value remain close to 1. Under this stronger requirement, every single error vector will approximately preserve its norm, and moreover all angles between different error vectors will be preserved. Since error information

backpropagates faithfully and isometrically through the network, this stronger requirement is called *dynamical isometry* [10].

A theoretical analysis of exact solutions to the nonlinear dynamics of learning in deep *linear* networks [10] revealed that weight initializations satisfying dynamical isometry yield a dramatic increase in learning speed compared to initializations that do not. For such linear networks, orthogonal weight initializations achieve dynamical isometry, and, remarkably, their learning time, measured in number of learning epochs, becomes *independent* of depth. In contrast, random Gaussian initializations do not achieve dynamical isometry, nor do they achieve depth-independent training times.

It remains unclear, however, how these results carry over to deep *nonlinear* networks. Indeed, empirically, a simple change from Gaussian to orthogonal initializations in nonlinear networks has yielded mixed results [11], raising important theoretical and practical questions. First, how does the entire distribution of singular values of a deep network's input-output Jacobian depend upon the depth, the statistics of random initial weights, and the shape of the nonlinearity? Second, what combinations of these ingredients can achieve dynamical isometry? And third, among the nonlinear networks that have neither vanishing nor exploding gradients, do those that in addition achieve dynamical isometry also achieve much faster learning compared to those that do not? Here we answer these three questions, and we provide a detailed summary of our results in the discussion.

## 2    Theoretical Results

In this section we derive expressions for the entire singular value density of the input-output Jacobian for a variety of nonlinear networks in the large-width limit. We compute the mean squared singular value of $\mathbf{J}$ (or, equivalently, the mean eiganvalue of $\mathbf{J}\mathbf{J}^T$), and deduce a rescaling that sets it equal to 1. We then examine two metrics that help quantify the conditioning of the Jacobian: $s_{\max}$, the maximum singular value of $\mathbf{J}$ (or, equivalently, $\lambda_{\max}$, the maximum eigenvalue of $\mathbf{J}\mathbf{J}^T$); and $\sigma^2_{JJ^T}$, the variance of the eigenvalue distribution of $\mathbf{J}\mathbf{J}^T$. If $\lambda_{\max} \gg 1$ and $\sigma^2_{JJ^T} \gg 1$ then the Jacobian is ill-conditioned and we expect the learning dynamics to be slow.

### 2.1    Problem setup

Consider an $L$-layer feed-forward neural network of width $N$ with synaptic weight matrices $\mathbf{W}^l \in \mathbb{R}^{N \times N}$, bias vectors $\mathbf{b}^l$, pre-activations $\mathbf{h}^l$ and post-activations $\mathbf{x}^l$, with $l = 1, \ldots, L$. The feed-forward dynamics of the network are governed by,

$$\mathbf{x}^l = \phi(\mathbf{h}^l)\,, \quad \mathbf{x}^l = \mathbf{W}^l \mathbf{h}^{l-1} + \mathbf{b}^l\,, \tag{1}$$

where $\phi : \mathbb{R} \to \mathbb{R}$ is a pointwise nonlinearity and the input is $\mathbf{h}^0 \in \mathbb{R}^N$. Now consider the input-output Jacobian $\mathbf{J} \in \mathbb{R}^{N \times N}$ given by

$$\mathbf{J} = \frac{\partial \mathbf{x}^L}{\partial \mathbf{h}^0} = \prod_{l=1}^{L} \mathbf{D}^l \mathbf{W}^l. \tag{2}$$

Here $\mathbf{D}^l$ is a diagonal matrix with entries $D^l_{ij} = \phi'(h^l_i)\,\delta_{ij}$. The input-output Jacobian $\mathbf{J}$ is closely related to the backpropagation operator mapping output errors to weight matrices at a given layer; if the former is well conditioned, then the latter tends to be well-conditioned for all weight layers. We therefore wish to understand the entire singular value spectrum of $\mathbf{J}$ for deep networks with randomly initialized weights and biases.

In particular, we will take the biases $b^l_i$ to be drawn i.i.d. from a zero mean Gaussian with standard deviation $\sigma_b$. For the weights, we will consider two random matrix ensembles: (1) random *Gaussian* weights in which each $W^l_{ij}$ is drawn i.i.d from a Gaussian with variance $\sigma^2_w/N$, and (2) random *orthogonal* weights, drawn from a uniform distribution over scaled orthogonal matrices obeying $(\mathbf{W}^l)^T \mathbf{W}^l = \sigma^2_w \mathbf{I}$.

### 2.2    Review of signal propagation

The random matrices $\mathbf{D}^l$ in eqn. (2) depend on the empirical distribution of pre-activations $\mathbf{h}^l$ entering the nonlinearity $\phi$ in eqn. (1). The propagation of this empirical distribution through different layers $l$

was studied in [12]. There, it was shown that in the large-$N$ limit this empirical distribution converges to a Gaussian with zero mean and variance $q^l$, where $q^l$ obeys a recursion relation induced by the dynamics in eqn. (1),

$$q^l = \sigma_w^2 \int \mathcal{D}h \, \phi \left( \sqrt{q^{l-1}}h \right)^2 + \sigma_b^2 \,, \tag{3}$$

with initial condition $q^0 = \frac{1}{N} \sum_{i=1}^{N} (h_i^0)^2$, and where $\mathcal{D}h = \frac{dh}{\sqrt{2\pi}} \exp\left(-\frac{h^2}{2}\right)$ denotes the standard Gaussian measure. This recursion has a fixed point obeying,

$$q^* = \sigma_w^2 \int \mathcal{D}h \, \phi \left( \sqrt{q^*}h \right)^2 + \sigma_b^2 \,. \tag{4}$$

If the input $\mathbf{h}^0$ is chosen so that $q^0 = q^*$, then we start at the fixed point, and the distribution of $\mathbf{D}^l$ becomes independent of $l$. Also, if we do not start at the fixed point, in many scenarios we rapidly approach it in a few layers (see [12]), so for large $L$, assuming $q^l = q^*$ at all depths $l$ is a good approximation in computing the spectrum of $\mathbf{J}$.

Another important quantity governing signal propagation through deep networks [12, 13] is

$$\chi = \frac{1}{N} \left\langle \mathrm{Tr} \left( \mathbf{DW} \right)^T \mathbf{DW} \right\rangle = \sigma_w^2 \int \mathcal{D}h \left[ \phi' \left( \sqrt{q^*}h \right) \right]^2 \,, \tag{5}$$

where $\phi'$ is the derivative of $\phi$. Here $\chi$ is the mean of the distribution of squared singular values of the matrix $\mathbf{DW}$, when the pre-activations are at their fixed point distribution with variance $q^*$. As shown in [12, 13] and Fig. 1, $\chi(\sigma_w, \sigma_b)$ separates the $(\sigma_w, \sigma_b)$ plane into two phases, chaotic and ordered, in which gradients exponentially explode or vanish respectively. Indeed, the mean squared singular value of $\mathbf{J}$ was shown simply to be $\chi^L$ in [12, 13], so $\chi = 1$ is a critical line of initializations with neither vanishing nor exploding gradients.

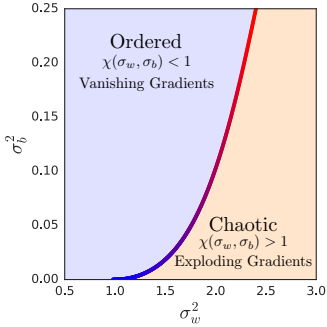

Figure 1: Order-chaos transition when $\phi(h) = \tanh(h)$. The critical line $\chi(\sigma_w, \sigma_b) = 1$ determines the boundary between two phases [12, 13]: (a) a chaotic phase when $\chi > 1$, where forward signal propagation expands and folds space in a chaotic manner and back-propagated gradients exponentially explode, and (b) an ordered phase when $\chi < 1$, where forward signal propagation contracts space in an ordered manner and back-propagated gradients exponentially vanish. The value of $q^*$ along the critical line separating the two phases is shown as a heatmap.

## 2.3 Free probability, random matrix theory and deep networks.

While the previous section revealed that the mean squared singular value of $\mathbf{J}$ is $\chi^L$, we would like to obtain more detailed information about the entire singular value distribution of $\mathbf{J}$, especially when $\chi = 1$. Since eqn. (2) consists of a product of random matrices, free probability [14, 15, 16] becomes relevant to deep learning as a powerful tool to compute the spectrum of $\mathbf{J}$, as we now review.

In general, given a random matrix $\mathbf{X}$, its limiting spectral density is defined as

$$\rho_X(\lambda) \equiv \left\langle \frac{1}{N} \sum_{i=1}^{N} \delta(\lambda - \lambda_i) \right\rangle_X \,, \tag{6}$$

where $\langle \cdot \rangle_X$ denotes the mean with respect to the distribution of the random matrix $\mathbf{X}$. Also,

$$G_X(z) \equiv \int_{\mathbb{R}} \frac{\rho_X(t)}{z - t} dt \,, \qquad z \in \mathbb{C} \setminus \mathbb{R} \,, \tag{7}$$

is the definition of the *Stieltjes transform* of $\rho_X$, which can be inverted using,

$$\rho_X(\lambda) = -\frac{1}{\pi} \lim_{\epsilon \to 0^+} \mathrm{Im} \, G_X(\lambda + i\epsilon) \,. \tag{8}$$

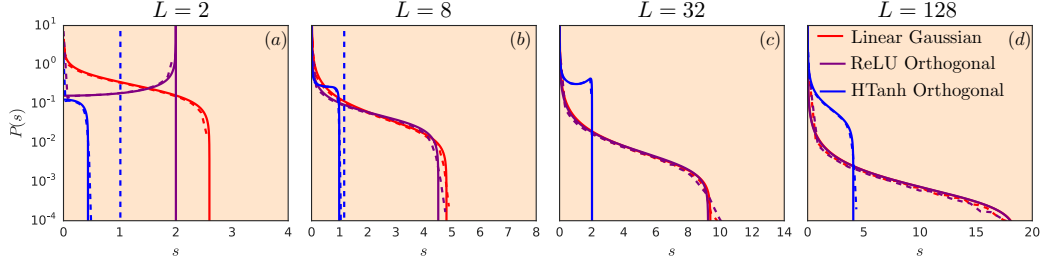

Figure 2: Examples of deep spectra at criticality for different nonlinearities at different depths. Excellent agreement is observed between empirical simulations of networks of width 1000 (dashed lines) and theoretical predictions (solid lines). ReLU and hard tanh are with orthogonal weights, and linear is with Gaussian weights. Gaussian linear and orthogonal ReLU have similarly-shaped distributions, especially for large depths, where poor conditioning and many large singular values are observed. On the other hand, orthogonal hard tanh is much better conditioned.

The Stieltjes transform $G_X$ is related to the moment generating function $M_X$,

$$M_X(z) \equiv zG_X(z) - 1 = \sum_{k=1}^{\infty} \frac{m_k}{z^k}, \tag{9}$$

where the $m_k$ is the $k$th moment of the distribution $\rho_X$, $m_k = \int d\lambda \, \rho_X(\lambda)\lambda^k = \frac{1}{N}\langle \text{tr}\mathbf{X}^k \rangle_X$. In turn, we denote the functional inverse of $M_X$ by $M_X^{-1}$, which by definition satisfies $M_X(M_X^{-1}(z)) = M_X^{-1}(M_X(z)) = z$. Finally, the *S-transform* [14, 15] is defined as,

$$S_X(z) = \frac{1+z}{zM_X^{-1}(z)}. \tag{10}$$

The utility of the S-transform arises from its behavior under multiplication. Specifically, if $\mathbf{A}$ and $\mathbf{B}$ are two freely-independent random matrices, then the S-transform of the product random matrix ensemble $\mathbf{AB}$ is simply the product of their S-transforms,

$$S_{AB}(z) = S_A(z)S_B(z). \tag{11}$$

Our first main result will be to use eqn. (11) to write down an implicit definition of the spectral density of $\mathbf{JJ}^T$. To do this we first note that (see Result 1 of the supplementary material),

$$S_{JJ^T} = \prod_{l=1}^{L} S_{W_l W_l^T} S_{D_l^2} = S_{WW^T}^L S_{D^2}^L, \tag{12}$$

where we have used the identical distribution of the weights to define $S_{WW^T} = S_{W_l W_l^T}$ for all $l$, and we have also used the fact the pre-activations are distributed independently of depth as $h_l \sim \mathcal{N}(0, q^*)$, which implies that $S_{D_l^2} = S_{D^2}$ for all $l$.

Eqn. (12) provides a method to compute the spectrum $\rho_{JJ^T}(\lambda)$. Starting from $\rho_{W^T W}(\lambda)$ and $\rho_{D^2}(\lambda)$, we compute their respective S-transforms through the sequence of equations eqns. (7), (9), and (10), take the product in eqn. (12), and then reverse the sequence of steps to go from $S_{JJ^T}$ to $\rho_{JJ^T}(\lambda)$ through the inverses of eqns. (10), (9), and (8). Thus we must calculate the S-transforms of $\mathbf{WW}^T$ and $\mathbf{D}^2$, which we attack next for specific nonlinearities and weight ensembles in the following sections. In principle, this procedure can be carried out numerically for an arbitrary choice of nonlinearity, but we postpone this investigation to future work.

## 2.4 Linear networks

As a warm-up, we first consider a linear network in which $\mathbf{J} = \prod_{l=1}^{L} \mathbf{W}^l$. Since criticality ($\chi = 1$ in eqn. (5)) implies $\sigma_w^2 = 1$ and eqn. (4) reduces to $q^* = \sigma_w^2 q^* + \sigma_b^2$, the only critical point is $(\sigma_w, \sigma_b) = (1, 0)$. The case of orthogonal weights is simple: $\mathbf{J}$ is also orthogonal, and all its singular values are 1, thereby achieving perfect dynamic isometry. Gaussian weights behave very differently.

The squared singular values $s_i^2$ of $\boldsymbol{J}$ equal the eigenvalues $\lambda_i$ of $\mathbf{JJ}^T$, which is a *product Wishart* matrix, whose spectral density was recently computed in [17]. The resulting singular value density of $\mathbf{J}$ is given by,

$$\rho(s(\phi)) = \frac{2}{\pi}\sqrt{\frac{\sin^3(\phi)\sin^{L-2}(L\phi)}{\sin^{L-1}((L+1)\phi)}}\,, \quad s(\phi) = \sqrt{\frac{\sin^{L+1}((L+1)\phi)}{\sin\phi\sin^L(L\phi)}}. \tag{13}$$

Fig. 2(a) demonstrates a match between this theoretical density and the empirical density obtained from numerical simulations of random linear networks. As the depth increases, this density becomes highly anisotropic, both concentrating about zero and developing an extended tail.

Note that $\phi = \pi/(L+1)$ corresponds to the minimum singular value $s_{\min} = 0$, while $\phi = 0$ corresponds to the maximum eigenvalue, $\lambda_{\max} = s_{\max}^2 = L^{-L}(L+1)^{L+1}$, which, for large $L$ scales as $\lambda_{\max} \sim eL$. Both eqn. (13) and the methods of Section 2.5 yield the variance of the eigenvalue distribution of $\mathbf{JJ}^T$ to be $\sigma_{JJ^T}^2 = L$. Thus for linear Gaussian networks, both $s_{\max}$ and $\sigma_{JJ^T}^2$ grow linearly with depth, signalling poor conditioning and the breakdown of dynamical isometry.

### 2.5 ReLU and hard-tanh networks

We first discuss the criticality conditions (finite $q^*$ in eqn. (4) and $\chi = 1$ in eqn. (5)) in these two nonlinear networks. For both networks, since the slope of the nonlinearity $\phi'(h)$ only takes the values 0 and 1, $\chi$ in eqn. (5) reduces to $\chi = \sigma_w^2 p(q^*)$ where $p(q^*)$ is the probability that a given neuron is in the linear regime with $\phi'(h) = 1$. As discussed above, we take the large-width limit in which the distribution of the pre-activations $h$ is a zero mean Gaussian with variance $q^*$. We therefore find that for ReLU, $p(q^*) = \frac{1}{2}$ is independent of $q^*$, whereas for hard-tanh, $p(q^*) = \int_{-1}^{1} dh \frac{e^{-h^2/2q^*}}{\sqrt{2\pi q^*}} = \mathrm{erf}(1/\sqrt{2q^*})$ depends on $q^*$. In particular, it approaches 1 as $q^* \to 0$.

Thus for ReLU, $\chi = 1$ if and only if $\sigma_w^2 = 2$, in which case eqn. (4) reduces to $q^* = \frac{1}{2}\sigma_w^2 q^* + \sigma_b^2$, implying that the only critical point is $(\sigma_w, \sigma_b) = (2, 0)$. For hard-tanh, in contrast, $\chi = \sigma_w^2 p(q^*)$, where $p(q^*)$ itself depends on $\sigma_w$ and $\sigma_b$ through eqn. (4), and so the criticality condition $\chi = 1$ yields a curve in the $(\sigma_w, \sigma_b)$ plane similar to that shown for the tanh network in Fig. 1. As one moves along this curve in the direction of decreasing $\sigma_w$, the curve approaches the point $(\sigma_w, \sigma_b) = (1, 0)$ with $q^*$ monotonically decreasing towards 0, i.e. $q^* \to 0$ as $\sigma_w \to 1$.

The critical ReLU network and the one parameter family of critical hard-tanh networks have neither vanishing nor exploding gradients, due to $\chi = 1$. Nevertheless, the entire singular value spectrum of $\mathbf{J}$ of these networks can behave very differently. From eqn. (12), this spectrum depends on the non-linearity $\phi(h)$ through $S_{D^2}$ in eqn. (10), which in turn only depends on the distribution of eigenvalues of $\mathbf{D}^2$, or equivalently, the distribution of squared derivatives $\phi'(h)^2$. As we have seen, this distribution is a Bernoulli distribution with parameter $p(q^*)$: $\rho_{D^2}(z) = (1 - p(q^*))\,\delta(z) + p(q^*)\,\delta(z-1)$. Inserting this distribution into the sequence eqn. (7), eqn. (9), eqn. (10) then yields

$$G_{D^2}(z) = \frac{1 - p(q^*)}{z} + \frac{p(q^*)}{z-1}\,, \qquad M_{D^2}(z) = \frac{p(q^*)}{z-1}\,, \qquad S_{D^2}(z) = \frac{z+1}{z+p(q^*)}\,. \tag{14}$$

To complete the calculation of $S_{JJ^T}$ in eqn. (12), we must also compute $S_{WW^T}$. We do this for Gaussian and orthogonal weights in the next two subsections.

#### 2.5.1 Gaussian weights

We re-derive the well-known expression for the $S$-transform of products of random Gaussian matrices with variance $\sigma_w^2$ in Example 3 of the supplementary material. The result is $S_{WW^T} = \sigma_w^{-2}(1+z)^{-1}$, which, when combined with eqn. (14) for $S_{D^2}$, eqn. (12) for $S_{JJ^T}$, and eqn. (10) for $M_X^{-1}(z)$, yields

$$S_{JJ^T}(z) = \sigma_w^{-2L}(z + p(q^*))^{-L}\,, \qquad M_{JJ^T}^{-1}(z) = \frac{z+1}{z}\big(z + p(q^*)\big)^L \sigma_w^{2L}. \tag{15}$$

Using eqn. (15) and eqn. (9), we can define a polynomial that the Stieltjes transform $G$ satisfies,

$$\sigma_w^{2L} G(Gz + p(q^*) - 1)^L - (Gz - 1) = 0\,. \tag{16}$$

The correct root of this equation is the one for which $G \sim 1/z$ as $z \to \infty$ [16]. From eqn. (8), the spectral density is obtained from the imaginary part of $G(\lambda + i\epsilon)$ as $\epsilon \to 0^+$.

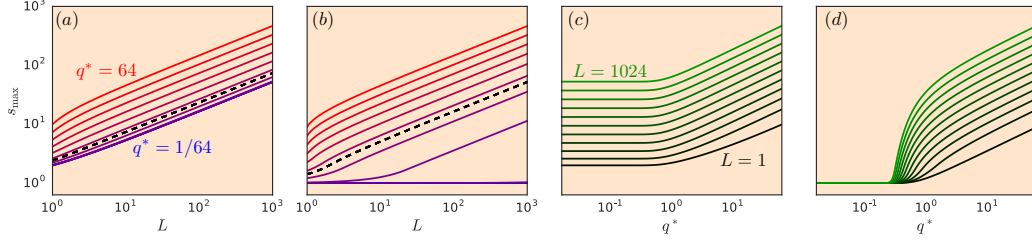

Figure 3: The max singular value $s_{\max}$ of $\mathbf{J}$ versus $L$ and $q^*$ for Gaussian (a,c) and orthogonal (b,d) weights, with ReLU (dashed) and hard-tanh (solid) networks. For Gaussian weights and for both ReLU and hard-tanh, $s_{\max}$ grows with $L$ for all $q^*$ (see a,c) as predicted in eqn. (17). In contrast, for orthogonal hard-tanh, but not orthogonal ReLU, at small enough $q^*$, $s_{\max}$ can remain $\mathcal{O}(1)$ even at large $L$ (see b,d) as predicted in eqn. (22). In essence, at fixed small $q^*$, if $p(q^*)$ is the large fraction of neurons in the linear regime, $s_{\max}$ only grows with $L$ after $L > p/(1-p)$ (see d). As $q^* \to 0$, $p(q^*) \to 1$ and the hard-tanh networks look linear. Thus the lowest curve in (a) corresponds to the prediction of linear Gaussian networks in eqn. (13), while the lowest curve in (b) is simply 1, corresponding to linear orthogonal networks.

The positions of the spectral edges, namely locations of the minimum and maximum eigenvalues of $\mathbf{JJ}^T$, can be deduced from the values of $z$ for which the imaginary part of the root of eqn. (16) vanishes, i.e. when the discriminant of the polynomial in eqn. (16) vanishes. After a detailed but unenlightening calculation, we find, for large $L$,

$$\lambda_{\max} = s_{\max}^2 = \left(\sigma_w^2 p(q^*)\right)^L \left(\frac{e}{p(q^*)} L + \mathcal{O}(1)\right).\tag{17}$$

Recalling that $\chi = \sigma_w^2 p(q^*)$, we find exponential growth in $\lambda_{\max}$ if $\chi > 1$ and exponential decay if $\chi < 1$. Moreover, *even at criticality when $\chi = 1$, $\lambda_{\max}$ still grows *linearly* with depth.

Next, we obtain the variance $\sigma_{JJ^T}^2$ of the eigenvalue density of $\mathbf{JJ}^T$ by computing its first two moments $m_1$ and $m_2$. We employ the Lagrange inversion theorem [18],

$$M_{JJ^T}(z) = \frac{m_1}{z} + \frac{m_2}{z^2} + \cdots, \qquad M_{JJ^T}^{-1}(z) = \frac{m_1}{z} + \frac{m_2}{m_1} + \cdots,\tag{18}$$

which relates the expansions of the moment generating function $M_{JJ^T}(z)$ and its functional inverse $M_{JJ^T}^{-1}(z)$. Substituting this expansion for $M_{JJ^T}^{-1}(z)$ into eqn. (15), expanding the right hand side, and equating the coefficients of $z$, we find,

$$m_1 = (\sigma_w^2 p(q^*))^L, \qquad m_2 = (\sigma_w^2 p(q^*))^{2L}\left(L + p(q^*)\right)/p(q^*).\tag{19}$$

Both moments generically either exponentially grow or vanish. However even at criticality, when $\chi = \sigma_w^2 p(q^*) = 1$, the variance $\sigma_{JJ^T}^2 = m_2 - m_1^2 = \frac{L}{p(q^*)}$ still exhibits linear growth with depth.

Note that $p(q^*)$ is the fraction of neurons operating in the linear regime, which is always less than 1. Thus for both ReLU and hard-tanh networks, no choice of Gaussian initialization can ever prevent this linear growth, both in $\sigma_{JJ^T}^2$ and $\lambda_{\max}$, implying that even critical Gaussian initializations will always lead to a failure of dynamical isometry at large depth for these networks.

### 2.5.2 Orthogonal weights

For orthogonal $\mathbf{W}$, we have $\mathbf{WW}^T = \mathbf{I}$, and the S-transform is $S_I = 1$ (see Example 2 of the supplementary material). After scaling by $\sigma_w$, we have $S_{WW^T} = S_{\sigma_w^2 I} = \sigma_w^{-2} S_I = \sigma_w^{-2}$. Combining this with eqn. (14) and eqn. (12) yields $S_{JJ^T}(z)$ and, through eqn. (10), yields $M_{JJ^T}^{-1}$:

$$S_{JJ^T}(z) = \sigma_w^{-2L}\left(\frac{z+1}{z+p(q^*)}\right)^L, \qquad M_{JJ^T}^{-1} = \frac{z+1}{z}\left(\frac{z+1}{z+p(q^*)}\right)^{-L}\sigma_w^{2L}.\tag{20}$$

Now, combining eqn. (20) and eqn. (9), we obtain a polynomial that the Stieltjes transform $G$ satisfies:

$$g^{2L}G(Gz + p(g) - 1)^L - (zG)^L(Gz - 1) = 0.\tag{21}$$

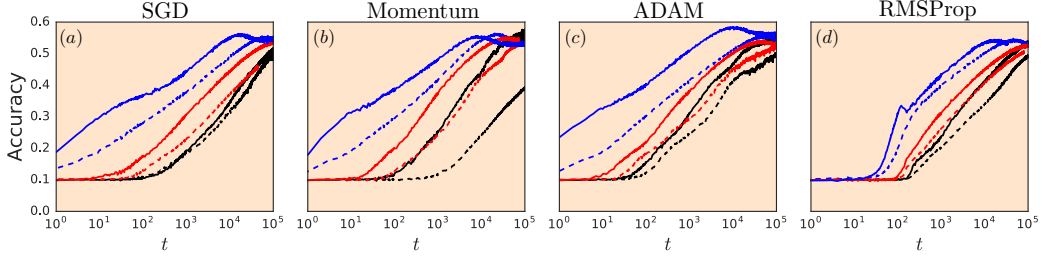

Figure 4: Learning dynamics, measured by *generalization* performance on a test set, for networks of depth 200 and width 400 trained on CIFAR10 with different optimizers. Blue is $tanh$ with $\sigma_w^2 = 1.05$, red is $tanh$ with $\sigma_w^2 = 2$, and black is ReLU with $\sigma_w^2 = 2$. Solid lines are orthogonal and dashed lines are Gaussian initialization. The relative ordering of curves robustly persists across optimizers, and is strongly correlated with the degree to which dynamical isometry is present at initialization, as measured by $s_{max}$ in Fig. 3. Networks with $s_{max}$ closer to 1 learn faster, even though all networks are initialized critically with $\chi = 1$. The most isometric orthogonal $tanh$ with small $\sigma_w^2$ trains several orders of magnitude faster than the least isometric ReLU network.

From this we can extract the eigenvalue and singular value density of $\mathbf{J}\mathbf{J}^T$ and $\mathbf{J}$, respectively, through eqn. (8). Figs. 2(b) and 2(c) demonstrate an excellent match between our theoretical predictions and numerical simulations of random networks. We find that at modest depths, the singular values are peaked near $\lambda_{max}$, but at larger depths, the distribution both accumulates mass at 0 and spreads out, developing a growing tail. Thus at *fixed* critical values of $\sigma_w$ and $\sigma_b$, both deep ReLU and hard-tanh networks have ill-conditioned Jacobians, even with orthogonal weight matrices.

As above, we can obtain the maximum eigenvalue of $\mathbf{J}\mathbf{J}^T$ by determining the values of $z$ for which the discriminant of the polynomial in eqn. (21) vanishes. This calculation yields,

$$\lambda_{max} = s_{max}^2 = \left(\sigma_w^2 p(q^*)\right)^L \frac{1 - p(q^*)}{p(q^*)} \frac{L^L}{(L-1)^{L-1}} . \tag{22}$$

For large $L$, $\lambda_{max}$ either exponentially explodes or decays, except at criticality when $\chi = \sigma_w^2 p(q^*) = 1$, where it behaves as $\lambda_{max} = \frac{1-p(q^*)}{p(q^*)}\left(eL - \frac{e}{2}\right) + \mathcal{O}(L^{-1})$. Also, as above, we can compute the variance $\sigma_{JJ^T}^2$ by expanding $M_{JJ^T}^{-1}$ in eqn. (20) and applying eqn. (18). At criticality, we find $\sigma_{JJ^T}^2 = \frac{1-p(q^*)}{p(q^*)}L$ for large $L$. Now the large $L$ asymptotic behavior of both $\lambda_{max}$ and $\sigma_{JJ^T}^2$ depends crucially on $p(q^*)$, the fraction of neurons in the linear regime.

For ReLU networks, $p(q^*) = 1/2$, and we see that $\lambda_{max}$ and $\sigma_{JJ^T}^2$ grow linearly with depth and dynamical isometry is unachievable in ReLU networks, even for critical orthogonal weights. In contrast, for hard tanh networks, $p(q^*) = \mathrm{erf}(1/\sqrt{2q^*})$. Therefore, one can always move along the critical line in the $(\sigma_w, \sigma_b)$ plane towards the point $(1, 0)$, thereby reducing $q^*$, increasing $p(q^*)$, and decreasing, to an arbitrarily small value, the prefactor $\frac{1-p(q^*)}{p(q^*)}$ controlling the linear growth of both $\lambda_{max}$ and $\sigma_{JJ^T}^2$. So unlike either ReLU networks, or Gaussian networks, one can achieve dynamical isometry up to depth $L$ by choosing $q^*$ small enough so that $p(q^*) \approx 1 - \frac{1}{L}$. In essence, this strategy increases the fraction of neurons operating in the linear regime, enabling orthogonal hard-tanh nets to mimic the successful dynamical isometry achieved by orthogonal linear nets. However, this strategy is unavailable for orthogonal ReLU networks. A demonstration of these results is shown in Fig. 3.

## 3 Experiments

Having established a theory of the entire singular value distribution of $\mathbf{J}$, and in particular of when dynamical isometry is present or not, we now provide empirical evidence that the presence or absence of this isometry can have a large impact on training speed. In our first experiment, summarized in Fig. 4, we compare three different classes of critical neural networks: (1) $tanh$ with small $\sigma_w^2 = 1.05$ and $\sigma_b^2 = 2.01 \times 10^{-5}$; (2) $tanh$ with large $\sigma_w^2 = 2$ and $\sigma_b^2 = 0.104$; and (3) ReLU with $\sigma_w^2 = 2$ and $\sigma_b^2 = 2.01 \times 10^{-5}$. In each case $\sigma_b$ is chosen appropriately to achieve critical initial conditions at the

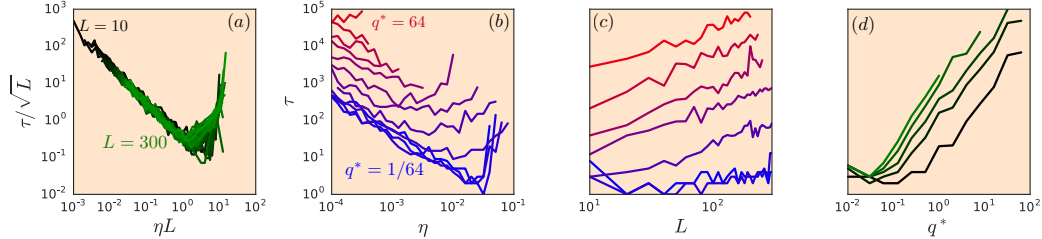

Figure 5: Empirical measurements of SGD training time $\tau$, defined as number of steps to reach $p \approx 0.25$ accuracy, for orthogonal $\mathtt{tanh}$ networks. In (a), curves reflect different depths $L$ at fixed small $q^* = 0.025$. Intriguingly, they all collapse onto a single universal curve when the learning rate $\eta$ is rescaled by $L$ and $\tau$ is rescaled by $1/\sqrt{L}$. This implies the optimal learning rate is $O(1/L)$, and remarkably, the optimal learning time $\tau$ grows only as $O(\sqrt{L})$. (b) Now different curves reflect different $q^*$ at fixed $L = 200$, revealing that smaller $q^*$, associated with increased dynamical isometry in $\mathbf{J}$, enables faster training times by allowing a larger optimal learning rate $\eta$. (c) $\tau$ as a function of $L$ for a few values of $q^*$. (d) $\tau$ as a function of $q^*$ for a few values of $L$. We see qualitative agreement of (c,d) with Fig. 3(b,d), suggesting a strong connection between $\tau$ and $s_{\max}$.

boundary between order and chaos [12, 13], with $\chi = 1$. All three of these networks have a mean squared singular value of $1$ with neither vanishing nor exploding gradients in the infinite width limit. These experiments therefore probe the specific effect of dynamical isometry, or the *entire shape* of the spectrum of $\mathbf{J}$, on learning. We also explore the degree to which more sophisticated optimizers can overcome poor initializations. We compare SGD, Momentum, RMSProp [19], and ADAM [20].

We train networks of depth $L = 200$ and width $N = 400$ for $10^5$ steps with a batch size of $10^3$. We additionally average our results over 30 different instantiations of the network to reduce noise. For each nonlinearity, initialization, and optimizer, we obtain the optimal learning rate through grid search. For SGD and SGD+Momentum we consider logarithmically spaced rates between $[10^{-4}, 10^{-1}]$ in steps $10^{0.1}$; for ADAM and RMSProp we explore the range $[10^{-7}, 10^{-4}]$ at the same step size. To choose the optimal learning rate we select a threshold accuracy $p$ and measure the first step when performance exceeds $p$. Our qualitative conclusions are fairly independent of $p$. Here we report results on a version of CIFAR10[1].

Based on our theory, we expect the performance advantage of orthogonal over Gaussian initializations to be significant in case (1) and somewhat negligible in cases (2) and (3). This prediction is verified in Fig. 4 (blue solid and dashed learning curves are well-separated, compared to red and black cases). Furthermore, the extent of dynamical isometry at initialization strongly predicts the speed of learning. The effect is large, with the most isometric case (orthogonal $\mathtt{tanh}$ with small $\sigma_w^2$) learning faster than the least isometric case (ReLU networks) by several orders of magnitude. Moreover, these conclusions robustly persist across all optimizers. Intriguingly, in the case where dynamical isometry helps the most ($\mathtt{tanh}$ with small $\sigma_w^2$), the effect of initialization (orthogonal versus Gaussian) has a much larger impact on learning speed than the choice of optimizer.

These insights suggest a more quantitative analysis of the relation between dynamical isometry and learning speed for orthogonal $\mathtt{tanh}$ networks, summarized in Fig. 5. We focus on SGD, given the lack of a strong dependence on optimizer. Intriguingly, Fig. 5(a) demonstrates the optimal training time is $O(\sqrt{L})$ and so grows *sublinearly* with depth $L$. Also Fig. 5(b) reveals that increased dynamical isometry enables faster training by making available larger (i.e. faster) learning rates. Finally, Fig. 5(c,d) and their similarity to Fig. 3(b,d) suggest a strong positive correlation between training time and max singular value of $\mathbf{J}$. Overall, these results suggest that dynamical isometry is correlated with learning speed, and controlling the *entire* distribution of Jacobian singular values may be an important design consideration in deep learning.

In Fig. 6, we explore the relationship between dynamical isometry and performance going *beyond* initialization by studying the evolution of singular values throughout training. We find that if dynamical isometry is present at initialization, it persists for some time into training. Intriguingly,

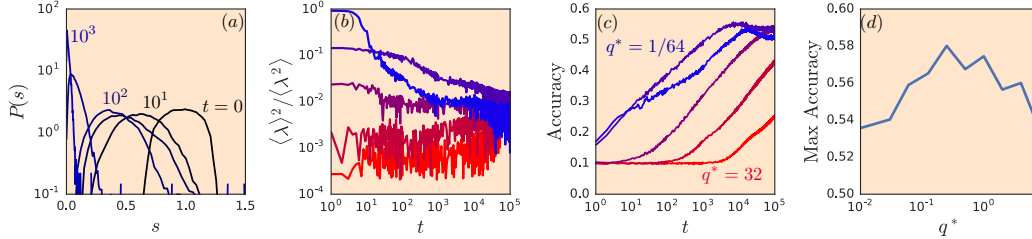

Figure 6: Singular value evolution of $\mathbf{J}$ for orthogonal $\mathtt{tanh}$ networks during SGD training. (a) The average distribution, over 30 networks with $q^* = 1/64$, at different SGD steps. (b) A measure of eigenvalue ill-conditioning of $\mathbf{JJ}^T$ ($\langle\lambda\rangle^2/\langle\lambda^2\rangle \leq 1$ with equality if and only if $\rho(\lambda) = \delta(\lambda - \lambda_0)$) over number of SGD steps for different initial $q^*$. Interestingly, the optimal $q^*$ that best maintains dynamical isometry in later stages of training is not simply the smallest $q^*$. (c) Test accuracy as a function of SGD step for those $q^*$ considered in (b). (d) Generalization accuracy as a function of initial $q^*$. Together (b,c,d) reveal that the optimal nonzero $q^*$, that best maintains dynamical isometry into training, *also* yields the fastest learning and best generalization accuracy.

perfect dynamical isometry at initialization ($q^* = 0$) is not the best choice for preserving isometry throughout training; instead, some small but nonzero value of $q^*$ appears optimal. Moreover, both learning speed and generalization accuracy peak at this nonzero value. These results bolster the relationship between dynamical isometry and performance beyond simply the initialization.

## 4 Discussion

In summary, we have employed free probability theory to analytically compute the entire distribution of Jacobian singular values as a function of depth, random initialization, and nonlinearity shape. This analytic computation yielded several insights into which combinations of these ingredients enable nonlinear deep networks to achieve dynamical isometry. In particular, deep linear Gaussian networks cannot; the maximum Jacobian singular value grows linearly with depth even if the second moment remains 1. The same is true for both orthogonal and Gaussian ReLU networks. Thus the ReLU nonlinearity destroys the dynamical isometry of orthogonal linear networks. In contrast, orthogonal, but not Gaussian, sigmoidal networks *can* achieve dynamical isometry; as the depth increases, the max singular value can remain $O(1)$ in the former case but grows linearly in the latter. Thus orthogonal sigmoidal networks rescue the failure of dynamical isometry in ReLU networks.

Correspondingly, we demonstrate, on CIFAR-10, that orthogonal sigmoidal networks can learn orders of magnitude faster than ReLU networks. This performance advantage is robust to the choice of a variety of optimizers, including SGD, momentum, RMSProp and ADAM. Orthogonal sigmoidal networks moreover have *sublinear* learning times with depth. While not as fast as orthogonal linear networks, which have depth independent training times [10], orthogonal sigmoidal networks have training times growing as the square root of depth. Finally, dynamical isometry, if present at initialization, persists for a large amount of time during training. Moreover, isometric initializations with longer persistence times yield both faster learning and better generalization.

Overall, these results yield the insight that the shape of the entire distribution of a deep network's Jacobian singular values can have a dramatic effect on learning speed; only controlling the second moment, to avoid exponentially vanishing and exploding gradients, can leave significant performance advantages on the table. Moreover, by pursuing the design principle of tightly concentrating the *entire* distribution around 1, we reveal that very deep feedfoward networks, with sigmoidal nonlinearities, can actually outperform ReLU networks, the most popular type of nonlinear deep network used today.

In future work, it would be interesting to extend our methods to other types of networks, including for example skip connections, or convolutional architectures. More generally, the performance advantage in learning that accompanies dynamical isometry suggests it may be interesting to explicitly optimize this property in reinforcement learning based searches over architectures [21].

### Acknowledgments

S.G. thanks the Simons, McKnight, James S. McDonnell, and Burroughs Wellcome Foundations and the Office of Naval Research for support.

## Footnotes

[1]We use the standard CIFAR10 dataset augmented with random flips and crops, and random saturation, brightness, and contrast perturbations

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
