[Supplementary Material]

# Supplemental Material
## Resurrecting the sigmoid in deep learning through dynamical isometry: theory and practice

## 1 Theoretical results

**Result 1.** *The S-transform for $JJ^T$ is given by,*

$$S_{JJ^T} = S_{WW^T}^L \prod_{l=1}^{L} S_{D_l^2}. \tag{S1}$$

*Proof.* First notice that, by eqn. (9), $M(z)$ and thus $S(z)$ depend only on the moments of the distribution. The moments, in turn, can be defined in terms of traces, which are invariant to cyclic permutations, i.e.,

$$\mathrm{tr}(A_1 A_2 \cdots A_m)^k = \mathrm{tr}(A_2 \cdots A_m A_1)^k. \tag{S2}$$

Therefore the S-transform is invariant to cyclic permutations. Define matrices $Q$ and $\tilde{Q}$,

$$Q_L \equiv JJ^T = (D_L W_L \cdots D_1 W_1)(D_L W_L \cdots D_1 W_1)^T \tag{S3}$$

$$\tilde{Q}_L \equiv (W_L^T D_L^T D_L W_L)(D_{L-1} W_{L-1} \cdots D_1 W_1)(D_{L-1} W_{L-1} \cdots D_1 W_1)^T \tag{S4}$$

$$= (W_L^T D_L^T D_L W_L) Q_{L-1}, \tag{S5}$$

which are related by a cyclic permutation. Therefore the above argument shows that their S-transforms are equal, i.e. $S_{Q_L} = S_{\tilde{Q}_L}$. Then eqn. (11) implies that,

$$S_{JJ^T} = S_{Q_L} = S_{W_L^T D_L^T D_L W_L} S_{Q_{L-1}} \tag{S6}$$

$$= S_{D_L^T D_L W_L W_L^T} S_{Q_{L-1}} \tag{S7}$$

$$= S_{D_L^2} S_{W_L W_L^T} S_{Q_{L-1}} \tag{S8}$$

$$= \prod_{l=1}^{L} S_{D_l^2} S_{W_l W_l^T} \tag{S9}$$

$$= S_{WW^T}^L \prod_{l=1}^{L} S_{D_l^2}, \tag{S10}$$

where the last line follows since each weight matrix is identically distributed. $\square$

**Example 1.** *Products of Gaussian random matrices with variance $\sigma_w^2$ have the S transform,*

$$S_{WW^T}(z) = \frac{1}{\sigma_w^2(1+z)}. \tag{S11}$$

*Proof.* It is well-known (see, e.g. [16]) that the moments of a Wishart are proportional to the Catalan numbers, i.e.,

$$m_k(WW^T) = \sigma_w^{2k} \frac{1}{k+1} \binom{2k}{k}, \tag{S12}$$

whose generating function is

$$M_{WW^T}(z) = \frac{1}{2}\left(-2 + \frac{z}{\sigma_w^2} - \sqrt{\frac{z}{\sigma_w^2}\left(\frac{z}{\sigma_w^2} - 4\right)}\right). \tag{S13}$$

It is straightforward to invert this function,

$$M_{WW^T}^{-1}(z) = \sigma_w^2 \frac{(1+z)^2}{z}, \tag{S14}$$

so that, using eqn. (10),

$$S_{WW^T}(z) = \frac{1}{\sigma_w^2(1+z)} \tag{S15}$$

as hypothesized. $\square$

**Example 2.** *The S-transform of the identity is given by $S_I = 1$.*

*Proof.* The moments of the identity are all equal to one, so we have,

$$M_I(z) = \sum_{k=1}^{\infty} \frac{1}{z^k} = \frac{1}{z-1} \,, \tag{S16}$$

whose inverse is,

$$M_I^{-1}(z) = \frac{1+z}{z} \,, \tag{S17}$$

so that,

$$S_I = 1 \,. \tag{S18}$$

$\square$