[Reviews · NeurIPS 2017]

Reviewer 1



This work extends prior work examining linear networks, showing the conditions under which orthogonal initialization makes deep networks well-conditioned, for the case of ReLU and hard-Tanh networks. They find that for ReLU nets, orthogonal initialization should (does) not improve conditioning, but for hard-Tanh it does improve conditioning. The question of orthogonal initialization has been of particular interest for RNNs, which suffer most from ill-conditioning (this work should perhaps be mentioned e.g. https://arxiv.org/pdf/1511.06464.pdf, https://arxiv.org/pdf/1504.00941.pdf ) and indeed anecdotal results seem to be mixed. I do not have sufficient background to follow the proofs, but I think the results would be of interest to the NIPS community. However, I'm disappointed that the authors chose to use such an unrealistic model for their experiments (L219). The model is not specified clearly in the paper or supplement, but it sounds like a 200-layer fully-connected network (judging from a comment in the supplement). Certainly this is not a reasonable network for MNIST or CIFAR. I would like to see comparisons on reasonable models for the dataset being presented. Final accuracy is not reported (hard to tell from the plots in the supplement). Detailed comments L99: lighting/lightning 2.3 "It appears that sigma ~ 4\epsilon L^-1" where is this shown? Figures plot distributions for particular values of L. L166 I think you're saying that \sigma^CRE / \sigma^(linear+Gaussian) -> 1 . Make that clear because for CRE linear networks it's well-conditioned. Also, what about the O(1)? I hope it's O(eps) or something because you are looking at a function with a range of [0, 1] so O(1) kills you.

Reviewer 2



The article is focused on the problem of understanding the learning dynamics of deep neural networks depending on both the activation functions used at the different layers and on the way the weights are initialized. It is mainly a theoretical paper with some experiments that confirm the theoretical study. The core of the contribution is made based on the random matrix theory. In the first Section, the paper describes the setup -- a deep neural network as a sequence of layers -- and also the tools that will be used to study their dynamics. The analysis mainly relies on the study of the singular values density of the jacobian matrix, this density being computed by a 4 step methods proposed in the article. Then this methodology is applied for 3 different architectures: linear networks, RELU networks and tanh networks, and different initialization of the weights -- Ginibre ensemble and Circular Real Ensemble. The theoretical results mainly show that tanh and linear networks may be trained efficiently -- particularly using orthogonal matrix initialization -- while RELU networks are poorly conditioned for the two initialization methods. The experiments on MNIST and CIFAR-10 confirm these results. First, I must say that the paper is a little bit out of my main research domain. Particularly, I am not familiar with random matrix theory. But, considering that I am not aware of this type of article, I find the paper well written: the focus is clear, the methodology is well described, references include many relevant prior works helping to understand the contribution, and the authors have also included an experimental section. Concerning the theoretical part of the paper, the proposed results seem correct, but would certainly need a deeper checking by an expert of the domain. Concerning the experimental section, the experiments confirm the theoretical results. But I must say that I am a little bit skeptical concerning the topic of the paper since all the demonstrations are focused on the simple deep neural network architectures with linear+bias transformations between the layers. It is a little bit surprising to see results on CIFAR-10 which is typically a computer vision benchmark where convolutional models are used, which is not the case here. So I have some doubt concerning the relevance of the paper w.r.t existing architectures that are now much more complex than sequences of layers. On another side, this paper is certainly a step toward the analysis of more complex architectures, and the author could argue on how difficult it could/will be to use the same tools (random matrix theory) to target for example convolutional networks.

Reviewer 3



The authors of this manuscript investigate the learning dynamics of supervised deep learning models with nonlinear activation functions. It is shown that the singular value distribution of the input-output Jacobian matrix is essential to the initialization of model parameters. This claim performs well on tanh networks but not ReLU networks. Majors: (1) The authors may investigate with their theory why the layer-wise pretraining can provide a good warm start in the model learning. (2) I also wonder whether this random matrix theory is applicable to generative models such as deep Boltzmann machine and deep belief net. (3) The presentation of the method section should be revised to improve its readability to general readers. Minors: The authors tend to use lowercase when uppercase is needed, such as fig, eqnt, relu, and terms in the references.